# The methylglyoxal pathway is a sink for glutathione in *Salmonella* experiencing oxidative stress

Sashi Kant[1], Lin Liu[1], Andres Vazquez-Torres[1,2]*

1 University of Colorado School of Medicine, Department of Immunology and Microbiology, Aurora, Colorado, United States of America, 2 Veterans Affairs, Eastern Colorado Health Care System, Aurora, Colorado, United States of America

* andres.vazquez-torres@cuanschutz.edu

**Data Availability Statement:** All relevant data are within the paper and its Supporting Information files.

**Funding:** These studies were supported by grants BX0002073 and IK6BX005384 from the U.S.

## Abstract

*Salmonella* suffer the cytotoxicity of reactive oxygen species generated by the phagocyte NADPH oxidase in the innate host response. Periplasmic superoxide dismutases, catalases and hydroperoxidases detoxify superoxide and hydrogen peroxide ($H_2O_2$) synthesized in the respiratory burst of phagocytic cells. Glutathione also helps *Salmonella* combat the phagocyte NADPH oxidase; however, the molecular mechanisms by which this low-molecular-weight thiol promotes resistance of *Salmonella* to oxidative stress are currently unknown. We report herein that *Salmonella* undergoing oxidative stress transcriptionally and functionally activate the methylglyoxal pathway that branches off from glycolysis. Activation of the methylglyoxal pathway consumes a substantial proportion of the glutathione reducing power in *Salmonella* following exposure to $H_2O_2$. The methylglyoxal pathway enables *Salmonella* to balance glucose utilization with aerobic respiratory outputs. *Salmonella* take advantage of the metabolic flexibility associated with the glutathione-consuming methylglyoxal pathway to resist reactive oxygen species generated by the enzymatic activity of the phagocyte NADPH oxidase in macrophages and mice. Taken together, glutathione fosters oxidative stress resistance in *Salmonella* against the antimicrobial actions of the phagocyte NADPH oxidase by promoting the methylglyoxal pathway, an offshoot metabolic adaptation of glycolysis.

## Author summary

Low-molecular-weight thiols such as the tripeptide glutathione are essential components of the antioxidant arsenal of phylogenetically diverse organisms. *Salmonella* undergoing peroxide stress suffer a dramatic diminution in the pool of reduced glutathione. However, drops in glutathione reducing power are not paralleled with a buildup of oxidized glutathione that would be expected by direct oxidation of the tripeptide by $H_2O_2$. The involvement of glutathione synthetase, but not glutathione oxidoreductase that reduces oxidized glutathione, casts further doubt on the role of this low-molecular-weight thiol as scavenger of reactive oxygen species in *Salmonella* pathogenesis. Our investigations herein show

Department of Veterans Affairs, and grants R01AI54959 and R01AI136520 from the Division of Intramural Research, National Institute of Allergy and Infectious Diseases. The funders had no role in study design, data collection and analysis, decision to publish, or preparation of the manuscript.

**Competing interests:** The authors have declared that no competing interests exist.

that *Salmonella* sustaining oxidative stress consume large amounts of glutathione in the methylglyoxal pathway that is activated as overflow metabolism is favored during periods of oxidative stress. The lactoylglutathione hydrolase activity of glyoxalase II in the methylglyoxal pathway allows *Salmonella* to grow in glucose, while simultaneously fostering aerobic respiration. In conclusion, the contribution of glutathione to the resistance of *Salmonella* to oxidative stress mostly stems from the electrophilic detoxification of aldehyde byproducts of metabolism rather than serving as a scavenger of reactive oxygen species via direct nucleophilic attack of $H_2O_2$ peroxo linkage.

## Introduction

The Gram-negative, facultative intracellular bacterial pathogen *Salmonella enterica* serovar Typhimurium is a common cause of gastroenteritis in immunocompetent individuals, and a life-threatening disseminated disease in immunosuppressed people who suffer from HIV or *Plasmodium* co-infections or who carry mutations in IL-12 signaling or phagocyte NADPH oxidase genes [1–6]. The phagocyte NADPH oxidase is one of the most important determinants of the innate host response of humans and experimental animals to nontyphoidal *Salmonella* infections [6–8]. Reactive oxygen species engendered in the respiratory burst of professional phagocytes through the activity of phagocyte NADPH oxidase enzymatic complexes damage cysteine and methionine residues, metal prosthetic groups, and DNA molecules [9]. *Salmonella* adapt to the antimicrobial activity of the phagocyte NADPH oxidase by engaging the vacuolar-remodeling actions of the *Salmonella* pathogenicity island-2 type III secretion system, the detoxifying activities of superoxide dismutases, catalases and hydroperoxidases, the nucleic acid-repairing actions of DNA recombination systems, and the chaperoning features of thioredoxin [10–19].

In addition to these antioxidant defenses, *Salmonella* remodel metabolism during their adaptation to ROS engendered by the host [20,21]. *Salmonella* experiencing oxidative stress undergo overflow metabolism, an adaptive response in which glycolysis and substrate-level phosphorylation are favored over aerobic respiration and oxidative phosphorylation [21,22]. Shifting from oxidative phosphorylation to fermentation and substrate-level phosphorylation still meets redox and energetic needs of the cell. However, substrate-level phosphorylation yields low energetic outputs compared to oxidative phosphorylation. Therefore, to satisfy energetic, biosynthetic and redox demands with overflow metabolism, *Salmonella* undergoing oxidative stress crank up glucose uptake and utilization. Toxic phosphosugars accumulate following enhanced carbon flow through glycolysis [23]. Bacteria recycle phosphate from toxic phosphosugar intermediates by converting dihydroxyacetone phosphate to pyruvate in sequential reactions that involve the S-lactoylglutathione conjugate and the D-lactate intermediate in the methylglyoxal pathway [23,24].

In the following investigations, we tested if the methylglyoxal pathway partakes in the adaptive response of *Salmonella* to oxidative stress. Our research demonstrates that *Salmonella* sustaining oxidative stress activate the methylglyoxal pathway, drawing glutathione redox buffering capacity in the process. The glutathione-consuming activity of the methylglyoxal pathway enables glucose utilization and aerobic respiration, while bolstering resistance of *Salmonella* to the antimicrobial actions associated with oxidative products generated by the phagocyte NADPH oxidase in the innate host response of macrophages and mice.

## Results

### Peroxide stress alters the redox potential of *Salmonella*

To successfully grow in acute models of infection that are dominated by the antimicrobial activity of the phagocyte NADPH oxidase, *Salmonella* switch on overflow metabolism and stimulate glycolytic activity [21,22]. To gain more insights into the metabolic adaptations of *Salmonella* following oxidative stress, we measured the effect of $H_2O_2$ on the redox potential of *Salmonella* grown in MOPS minimum medium containing glucose as sole carbon source. These investigations showed that $H_2O_2$ treatment oxidizes the cytoplasm of *Salmonella*, as indicated by increased ratiometric roGFP measurements (**Fig 1A**). RoGFP fluorescence reflects the GSH/GSSG redox buffering capacity of the cell [25]. In agreement with the roGFP estimations, $H_2O_2$-treated *Salmonella* suffered a decrease in the GSH/GSSG ratio (**Fig 1B**). Drops in the GSH/GSSG redox power following treatment of *Salmonella* with $H_2O_2$ were explained by the diminution in the concentration of GSH. However, drops in GSH were not reciprocated by the gains in oxidized GSH (GSSG) that would be expected via the targeted oxidation of GSH by $H_2O_2$ (**Fig 1B**). These observations are consistent with previous published work that reported that the concentration of GSH diminishes in $H_2O_2$-treated *Salmonella* without corresponding increases in GSSG [14]. Together, these investigations indicate that the disappearance of GSH in *Salmonella* undergoing peroxide stress does not seem to reflect the oxidation of this low-molecular-weight thiol by direct nucleophilic attack to the peroxo linkage of $H_2O_2$.

### *Salmonella* undergoing oxidative stress utilize the methylglyoxal pathway

Based on the above observations, we turned our attention to pathways that consume GSH, and focused on the methylglyoxal pathway. We hypothesized that the loss of GSH in *Salmonella*

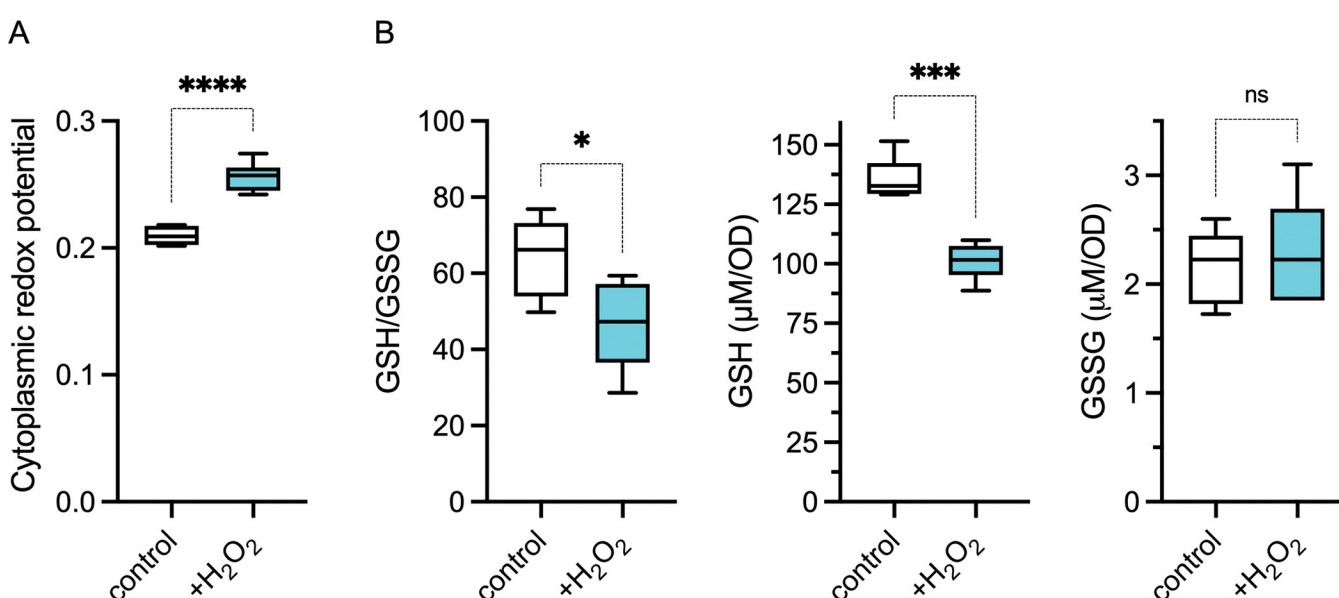

**Fig 1. Redox potential in *Salmonella* experiencing peroxide stress.** (A) Cytoplasmic redox potential was estimated by recording the fluorescent spectra of roGFP2 in bacteria grown in MOPS-GLC medium to $OD_{600}$ of 0.25. Some of the bacterial cultures were treated for 1 min with 400 μM $H_2O_2$. The relative redox potential denotes the ratio of roGFP2$_{ox}$ /roGFP2$_{red}$ emission signals at 510 nm after excitation at 405 and 480 nm. Data are the mean ± S.D. (N = 6). ****, $p< 0.0001$, as determined by unpaired *t*-test. (B) Estimations of reduced and oxidized glutathione (GSH and GSSG, respectively) in *Salmonella* strains grown to an $OD_{600}$ of 0.25 in MOPS-GLC minimal medium, pH 7.2. Where indicated, samples were treated with 400 μM $H_2O_2$ for 30 min prior to glutathione measurements. Data are the mean ± S.D. (N = 5) *, ***, $p < 0.05$ and $p < 0.001$, respectively as determined by unpaired *t*-test.

experiencing peroxide stress may reflect the utilization of this tripeptide in the methylglyoxal pathway. The methylglyoxal pathway, which branches off from glycolysis, generates the electrophile methylglyoxal during the turnover of glycolytic phosphosugar intermediates. Reaction of methylglyoxal with GSH forms the S-lactoylglutathione conjugate (**Figs 2 and A in S1 Text**). To begin testing if peroxide induces the methylglyoxal pathway in *Salmonella*, we

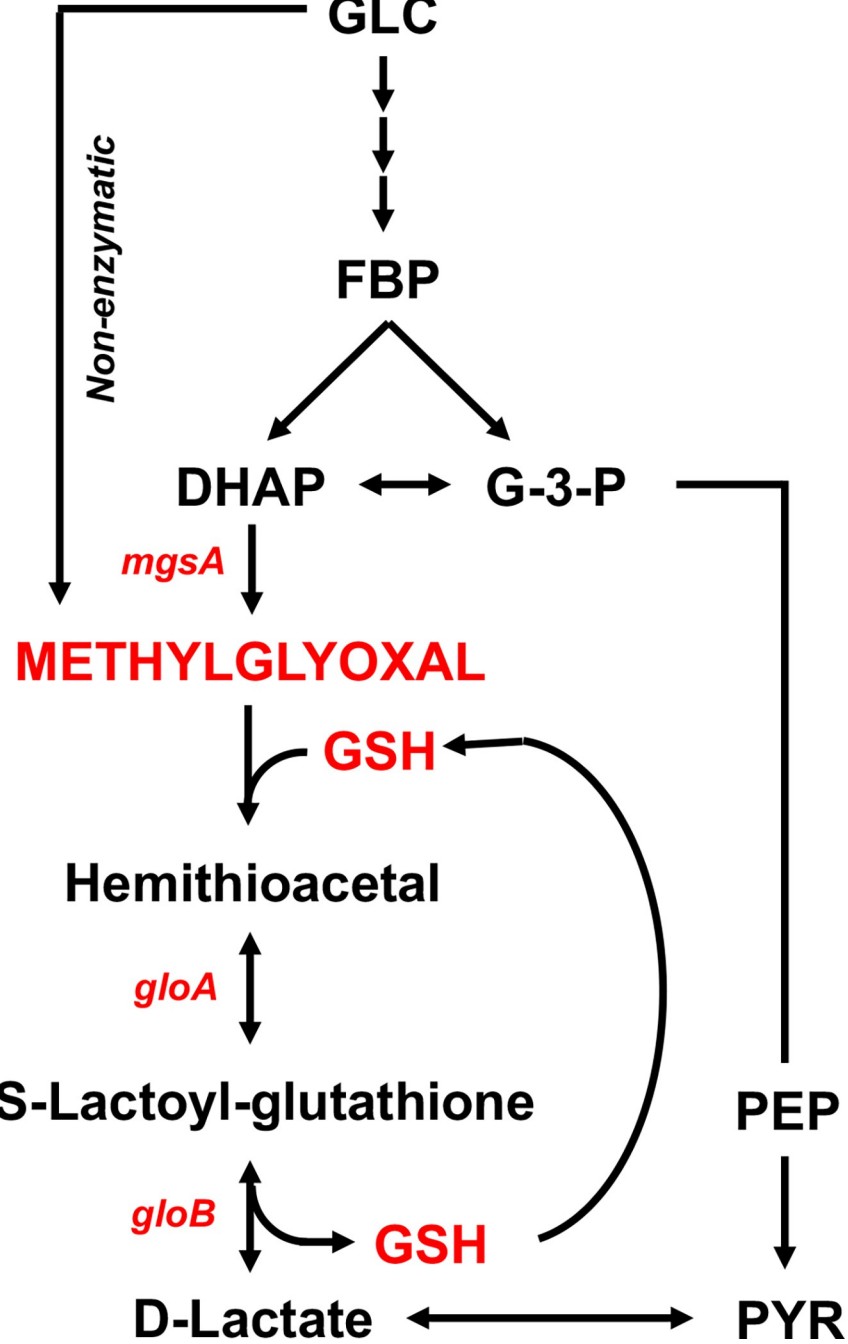

**Fig 2. Schematic representation of methylglyoxal pathway in *Salmonella*.** GLC, Glucose; FBP, Fructose 1,6 bis phosphate; DHAP, Dihydroxyacetone phosphate; G-3P, Glyceraldehyde 3-phosphate; PEP, Phosphoenolpyruvate; PYR, Pyruvate.

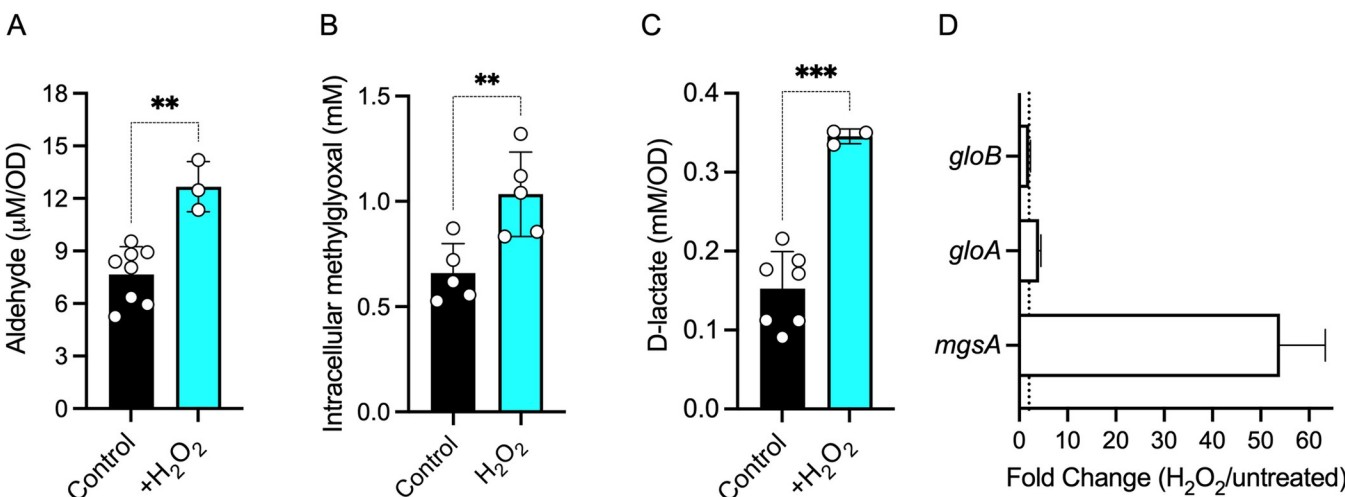

**Fig 3. Oxidative stress stimulates utilization of the methylglyoxal pathway.** Intracellular (A,B) aldehyde and (C) D-lactate concentrations in *Salmonella* grown to an $OD_{600}$ of 0.25 in MOPS-GLC minimal medium, pH 7.2. Selected samples were treated with 400 μM $H_2O_2$ for 30 min. Data are the mean ± S.D. (N = 3–8). **, ***, $p < 0.01$ and $p < 0.001$, respectively as determined by unpaired *t*-test. (D) Gene expression analysis quantified by qRT-PCR of specimens isolated from *Salmonella* grown to an $OD_{600}$ of 0.25 in MOPS-GLC minimal medium. Where indicated, bacterial cultures were treated with 400 μM $H_2O_2$ for 30 min before the RNA was isolated. The data, normalized to the *rpoD* housekeeping gene, represent the average fold-change ± S.D (N = 3). Dashed lines depict the 2-fold upregulated marks.

quantified the levels of aldehyde and D-lactate in control and $H_2O_2$-treated *Salmonella* grown to log phase in MOPS-GLC minimum medium. Treatment of *Salmonella* with $H_2O_2$ resulted in significant increases in the concentrations of aldehyde ($p < 0.01$) (**Fig 3A**) and D-lactate ($p < 0.001$) (**Fig 3B**). These findings indicate that *Salmonella* activate the methylglyoxal pathway following $H_2O_2$ treatment, providing a reasonable explanation for the observed decreases of GSH without the corresponding gains in GSSG in $H_2O_2$-treated S*almonella*. The biochemical findings were supported by targeted transcriptional analyses that showed the selective induction of methylglyoxal *mgsA*, *gloA* and *gloB* genes in *Salmonella* following treatment with $H_2O_2$ (**Figs 3C and B in S1 Text**).

## The methylglyoxal pathway supports growth of *Salmonella* on glucose

Having established the activation of the methylglyoxal pathway in *Salmonella* following oxidative stress, we determined if this pathway participates in the growth of *Salmonella* in glycolytic substrates. Because the methylglyoxal pathway is an offshoot of glycolysis, we monitored growth of methylglyoxal mutants in glucose medium. These investigations revealed that Δ*gloB Salmonella* deficient in glyoxalase II grow poorly in MOPS-GLC minimum medium (**Fig 4A**). Expression of the *gloB* gene under its native promoter from the low copy plasmid pWSK29 complemented the defective growth of Δ*gloB Salmonella* in MOPS-GLC minimum medium (**Fig C in S1 Text**). In contrast to Δ*gloB Salmonella*, isogenic controls lacking the *msgA* or *gloA* genes that encode methylglyoxal synthase or glyoxalase I, respectively, grew in MOPS-GLC minimum medium as effectively as wildtype cells (**Fig 4A**). These findings indicate that recycling of GSH from the S-lactoylglutathione intermediate by the *gloB*-encoded glyoxalase II is critical for the growth of *Salmonella* on glucose. As predicted by the enzymatic reaction catalyzed by glyoxalase II, the Δ*gloB* strain had a diminished GSH/GSSG buffering capacity, which was associated with low GSH concentrations compared to wildtype controls ($p < 0.0001$) (**Fig 4B**). Nonetheless, wildtype and Δ*gloB Salmonella* contained similar concentrations of GSSG (**Fig 4B**). Together, these findings suggest that the inability of the Δ*gloB* mutant to recycle GSH contributes to its poor utilization of glucose.

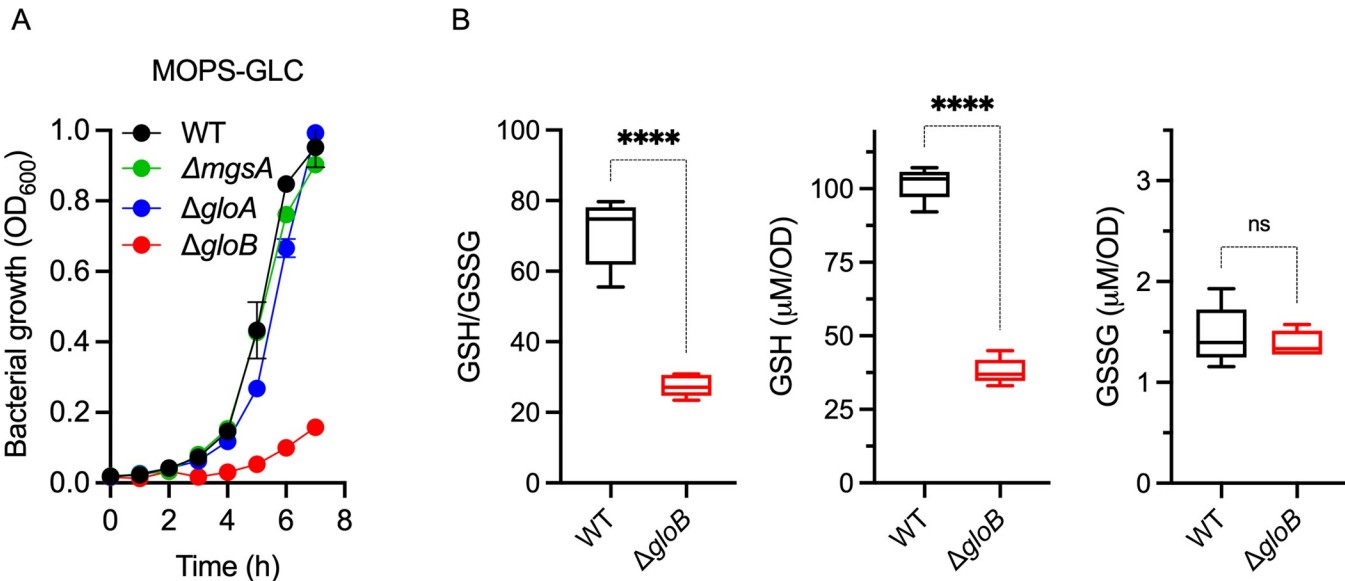

**Fig 4. Glyoxalase-II is important for growth of *Salmonella* strains in glucose.** (A) Aerobic growth of *Salmonella* in MOPS-GLC minimum medium, pH 7.2, at 37°C in a shaking incubator as assessed by $OD_{600}$. Data are the mean ± S.D (N = 3). (B) The content of reduced and oxidized glutathione (GSH and GSSG, respectively) was measured in wildtype and Δ*gloB Salmonella* strains grown to an $OD_{600}$ of 0.25 in MOPS-GLC minimum medium, pH 7.2. Data are the mean ± S.D. (N = 5) **, ****, $p < 0.01$ and $p < 0.0001$, respectively, as determined by unpaired *t*-test.

## The methylglyoxal pathway facilitates resistance of *Salmonella* to peroxide killing

We next examined the importance of the methylglyoxal pathway in resistance of *Salmonella* to $H_2O_2$ killing. We first tested the susceptibility of *Salmonella* grown to stationary phase in LB broth when exposed to $H_2O_2$ in PBS. Strains of *Salmonella* bearing Δ*msgA*, Δ*gloA* or Δ*gloB* deletion alleles were hypersusceptible ($p < 0.0001$) to $H_2O_2$ (**Fig D in S1 Text**), demonstrating that the methylglyoxal pathway is important in the metabolic adaptation of *Salmonella* to oxidative stress. The methylglyoxal mutants grown in MOPS-GLC or EG minimal media were also hypersusceptible to the bactericidal and bacteriostatic activities of $H_2O_2$ (**Figs 5A and D in S1 Text**). As shown above, exposure of wildtype *Salmonella* to $H_2O_2$ diminished the GSH/GSSG ratio, which reflected drops in GSH but not GSSG (**Fig 5B**). In comparison, the basal GSH/GSSG ratio was lower in Δ*gloA* and Δ*gloB Salmonella* strains, due to a mark reduction in the concentration of GSH. Interestingly, the addition of $H_2O_2$ neither dropped the GSH/GSSG ratio nor the basal concentrations of GSH in Δ*gloB Salmonella* (**Figs 5B and E in S1 Text**).

We next examined whether the methylglyoxal pathway help *Salmonella* resist oxidative stress when challenged in a carbon source other than glucose. For these studies we chose Casamino acids, which enter metabolism at lower steps of glycolysis and at various steps of the TCA. Moreover, metabolism of Casamino acids relies more heavily on oxidative phosphorylation than substrate-level phosphorylation [26]. Methylglyoxal pathway mutants displayed similar resistance to $H_2O_2$ in MOPS-Casamino acids or E-salts Casamino acids media compared to wildtype *Salmonella* controls (**Figs 5C and D in S1 Text**). Wildtype and Δ*gloB Salmonella* grown and exposed to $H_2O_2$ in MOPS-CAA minimum medium also showed similar levels of GSH and GSSG (**Fig 5D**). Of note, *Salmonella* growing in Casamino acid medium did not suffer decreases in the intracellular concentration of GSH after exposure to 400 μM $H_2O_2$.

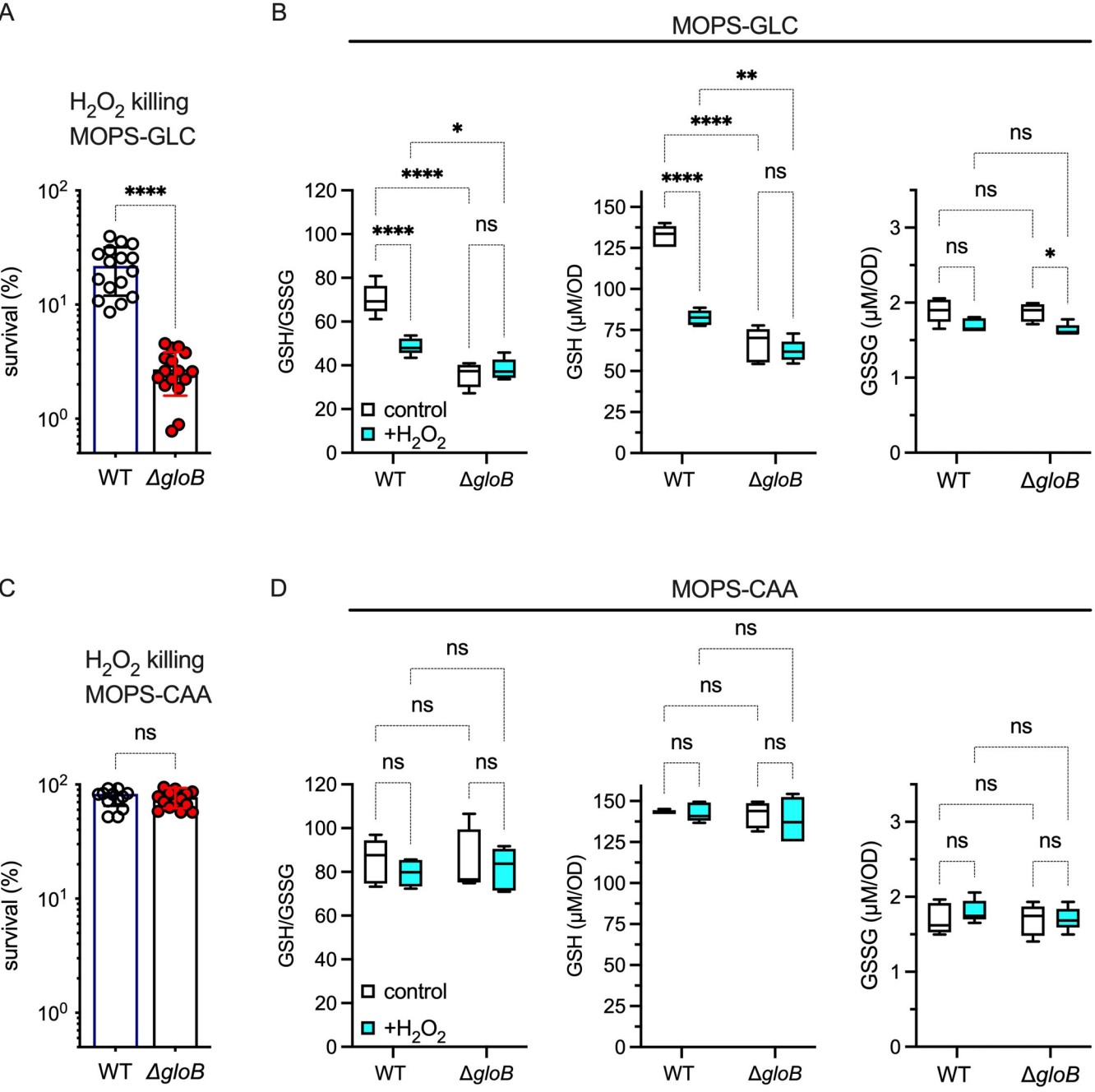

**Fig 5. Importance of the methylglyoxal pathway in resistance of *Salmonella* to peroxide killing.** (A) Bacterial cultures were grown overnight in LB broth, diluted to $2\times10^5$ CFU/ml in PBS and treated for 2h with 400 μM $H_2O_2$. Killing, expressed as percent survival compared to the bacterial burden at time zero, is the mean ± S.D. (N = 16); $p<0.0001$ as determined by unpaired *t*-test. (B). The GSH/GSSG content was estimated as described in Fig 1. Where indicated, bacterial cells grown in MOPS-GLC minimum medium to $OD_{600}$ of 0.25 were treated with 400 μM $H_2O_2$ for 30 min. Data are the mean ± S.D. (N = 5) *, **, ****, $p < 0.05$, $p < 0.01$ and $p < 0.0001$, respectively as determined by two-way ANOVA.

Cumulatively, these investigations suggest that the bulk of GSH that is consumed in *Salmonella* undergoing peroxide stress in glucose-based media reflects the electrophilic reaction of GSH with an aldehyde intermediate generated in the methylglyoxal pathway, and not the direct nucleophilic attack of the thiolate group in this low-molecular-weight thiol towards $H_2O_2$.

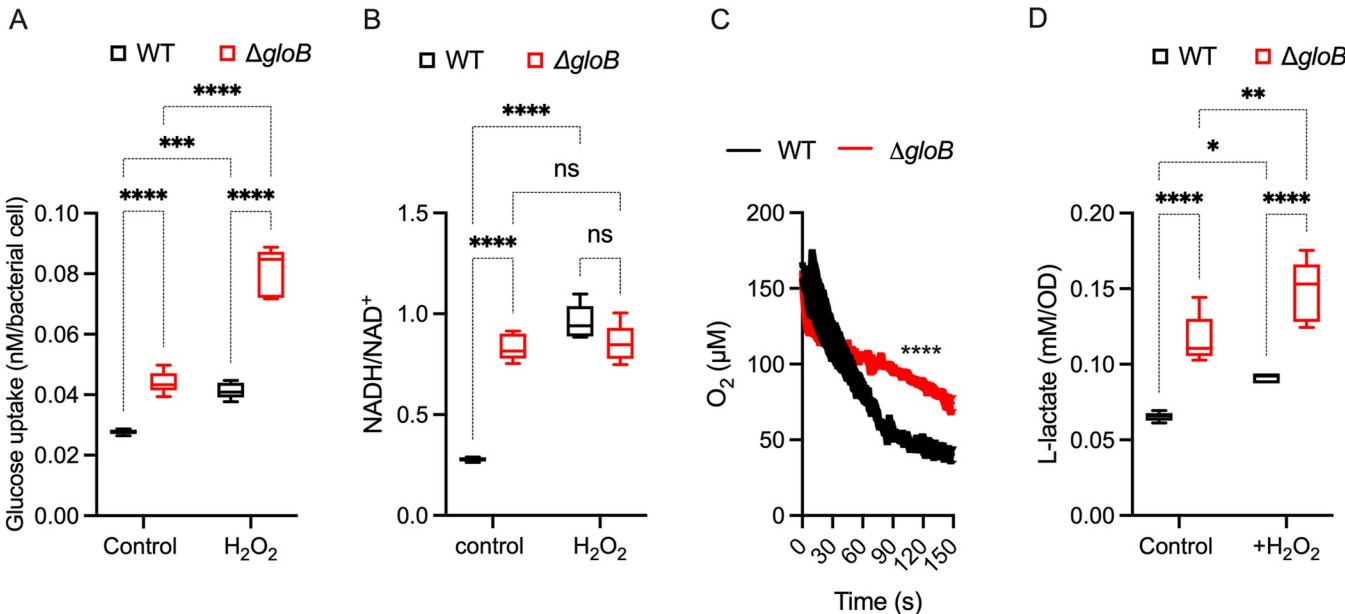

**Fig 6. The methylglyoxal pathway facilitates glucose and energy metabolism in *Salmonella* experiencing peroxide stress.** (A) Intracellular concentrations of glucose and (B) reduced and oxidized nicotinamide adenine nucleotide and (D) L-lactate in *Salmonella* grown aerobically in MOPS-GLC minimum medium, pH 7.2 to an $OD_{600}$ of 0.25 at 37°C. Where indicated, bacterial cultures were treated with 400 μM $H_2O_2$ for 30 min. Data are the mean ± S.D. (N = 5–6). ***, ****, $p < 0.001$ and $p < 0.0001$, respectively, as determined by two-way ANOVA. (C) Consumption of $O_2$ by log phase *Salmonella* grown in MOPS-GLC minimal medium, pH 7.2, was measured polarographically in an APOLLO-4000 free radical analyzer equipped with an ISO-OXY O2 probe. Data are the mean ± S.D (N = 7–10). **** $p<0.0001$ as determined by unpaired *t*-test.

## The methylglyoxal pathway promotes aerobic respiration

We explored in more detail the molecular mechanisms by which the methylglyoxal pathway promotes growth of *Salmonella* in glucose and resistance to peroxide stress. Glucose utilization and redox balance are excellent predictors of the adaptation of *Salmonella* to oxidative stress [21,22]. Consistent with previous work [22], wildtype *Salmonella* exposed to $H_2O_2$ increased glucose uptake (**Fig 6A**). To our surprise, the Δ*gloB* strain supported higher uptake of glucose than wildtype *Salmonella* grown in MOPS-GLC minimum medium (**Fig 6A**). The addition of $H_2O_2$ further increased glucose uptake in Δ*gloB Salmonella* (**Fig 6A**). Compared to wildtype controls grown in MOPS-GLC minimum medium, the basal NADH/NAD$^+$ ratio was higher in the Δ*gloB* mutant (**Fig 6B**). The high NADH/NAD$^+$ ratio in Δ*gloB Salmonella* reflected both higher levels of NADH and lower concentrations of NAD$^+$ compared to wildtype controls (**Fig F in S1 Text**). Moreover, the Δ*gloB* mutant supported low aerobic respiration compared to wildtype controls grown in MOPS-GLC minimum medium (**Fig 6C**). The low aerobic respiration and high glucose uptake recorded in Δ*gloB Salmonella* may have contributed to the NADH/NAD$^+$ ratios recorded in this strain, as NADH dehydrogenases in the electron transport chain and the glycolytic glyceraldehyde 3P-dehydrogenase are sizable sources of NADH oxidation and NAD$^+$ utilization, respectively. As expected by the glycolytic switch recorded in *Salmonella* undergoing peroxide stress [21,22], the addition of $H_2O_2$ to cultures of wildtype *Salmonella* increased the NADH/NAD$^+$ ratio (**Fig 6B**). $H_2O_2$ treatment did not change the already high NADH/NAD$^+$ ratio recorded in Δ*gloB Salmonella* growing in glucose-based media.

### Resistance of *Salmonella* to products of the phagocyte NADPH oxidase involves the methylglyoxal pathway

Having established roles for the *gloB*-encoded glyoxalase-II on the utilization of glucose and the resistance of *Salmonella* to $H_2O_2$, we tested the virulence of mutants in the methylglyoxal pathway in both bone-marrow-derived macrophages and a chronic granulomatous disease model of *Salmonella* infection dominated by the antimicrobial activity of the phagocyte NADPH oxidase. Compared to wildtype isogenic controls, *Salmonella* bearing deletions in *mgsA*, *gloA* or *gloB* genes were hypersusceptible ($p < 0.0001$) to the antimicrobial activity of bone-marrow-derived macrophages isolated from C57BL/6 mice (**Fig 7A**). The hypersusceptibility of these methylglyoxal mutants was already evident after 2 h of infection, a time when the activity of phagocyte NADPH oxidase is maximal [27]. These findings demonstrate that *Salmonella* deficient in the methylglyoxal pathway are at a fitness disadvantage during a time when macrophages produce oxidative products in the respiratory burst. In support of this idea, methylglyoxal pathway mutants were as virulent as a wildtype control in bone-marrow-derived macrophages isolated from *Cybb*$^{-/-}$ mice, which lack the gp91*phox* subunit of phagocyte NADPH oxidase (**Fig 7A**). The importance of the methylglyoxal pathway in the resistance of *Salmonella* to the respiratory burst of the innate host response was tested in C57BL/6 mice after i.p. inoculation of equal numbers of wildtype and Δ*gloB Salmonella*. Quantification of bacterial burdens in livers and spleens of infected mice 3 days after challenge showed lower burdens of Δ*gloB* mutant *Salmonella* in livers ($p < 0.0001$) and spleens ($p < 0.001$) compared to wildtype controls (**Fig 7B and 7C**). Importantly, Δ*gloB Salmonella* regained virulence in *Cybb*$^{-/-}$ mice deficient in the gp91*phox* membrane-bound subunit of phagocyte NADPH oxidase (**Fig 7B and 7C**). These data demonstrate that the recycling of GSH through the cleavage of S-lactoylglutathione contributes to the antioxidant defenses of *Salmonella* and the resistance of this intracellular pathogen to the antimicrobial activity of the phagocyte NADPH oxidase during the innate host response of macrophages and mice.

## Discussion

Our investigations have demonstrated a critical role for the methylglyoxal pathway in resistance of *Salmonella* to oxidative killing. Peroxide stress activates transcription of methylglyoxal pathway genes and induces the accumulation of the methylglyoxal intermediates. Our research demonstrates that the methylglyoxal pathway boosts the resistance of *Salmonella* to peroxide killing, while protecting intracellular *Salmonella* from the toxicity emanating from the phagocyte NADPH oxidase in macrophages and an acute model of salmonellosis.

The partial suppression of oxidative phosphorylation in *Salmonella* undergoing oxidative stress creates an energetic and redox balancing dilemma. *Salmonella* resolve this conundrum by increasing glucose utilization [21,22], which generates ATP in lower glycolysis and the fermentation of pyruvate to acetate. Increased glycolytic activity also boosts NADPH synthesis in pentose phosphate pathway, fueling antioxidant defenses such as glutathione and thioredoxin reductases. Moreover, by redirecting carbon flow to lactate fermentation and the reductive branch of TCA, overflow metabolism supplements the shrinking redox balancing capacity of an oxidized electron transport chain [21]. To maintain energetic and redox outputs with the low efficiency of glycolysis and fermentation, bacteria are forced to liberate inorganic phosphate from phosphosugar intermediates arising as glycolysis is cranked up. Our data show that *Salmonella* solve this predicament by upregulating the methylglyoxal pathway. Accordingly, *Salmonella* undergoing oxidative stress not only transcribe genes in the methylglyoxal pathway, but also generate aldehyde intermediates and activate production of D-lactate.

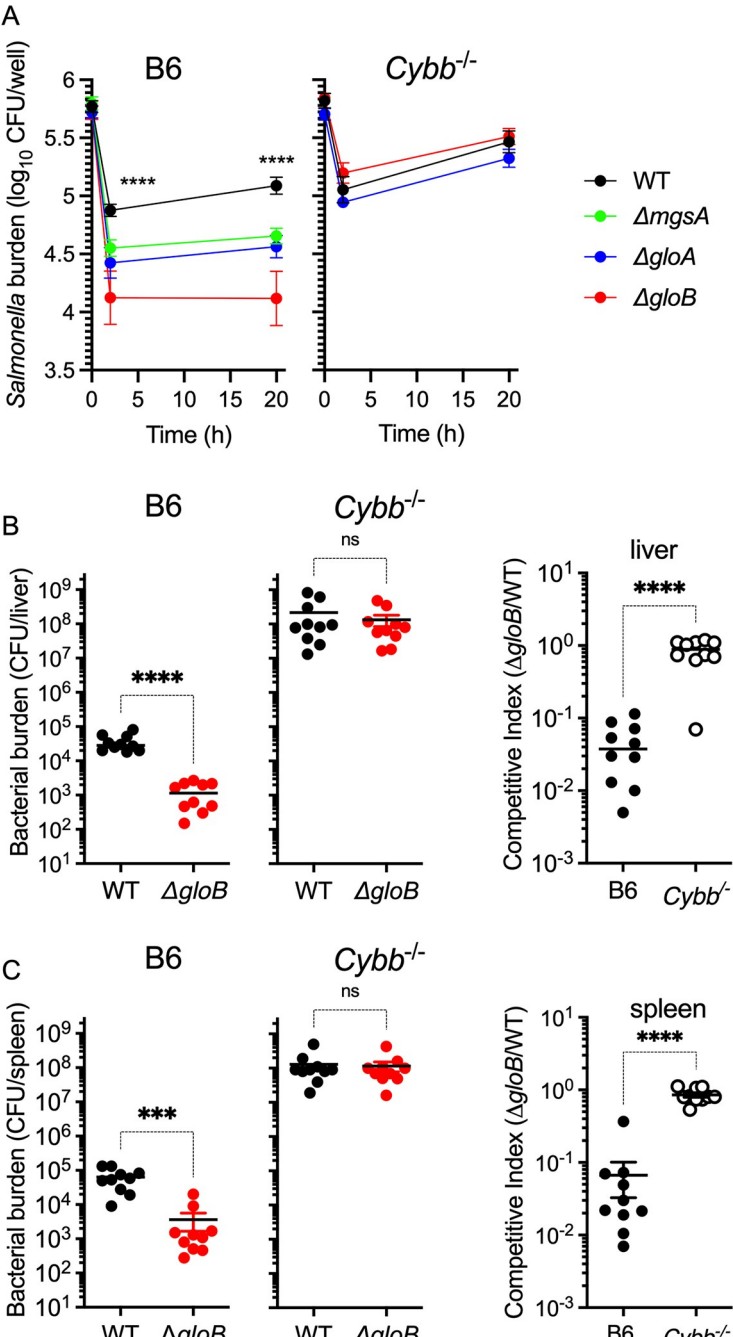

**Fig 7. Role of the methylglyoxal pathway in resistance of *Salmonella* to phagocyte NADPH oxidase.** (A) Bacterial burden in bone-marrow-derived macrophages isolated from C57BL/6 and *Cybb*$^{-/-}$ mice. Bacterial burden and competitive index of *Salmonella* strains in (B) livers and (C) spleens of C57BL/6 and *Cybb*$^{-/-}$ mice after i.p. inoculation with 100 CFU of equal numbers of wildtype and Δ*gloB Salmonella* (N = 10). Statistical differences (***, ****, $p < 0.001$ and $p < 0.0001$, respectively) were calculated by unpaired *t*-test.

Detoxification of phosphosugars generates the electrophilic intermediate methylglyoxal, an aldehyde responsible for the formation of advance glycation end-products [28]. Methylglyoxal is detoxified by its spontaneous reaction GSH or via the enzymatic activity of methylglyoxal synthase. Ironically, GSH is not consumed by the nucleophilic avidity that the tripeptide has

for $H_2O_2$, but rather expended indirectly via its electrophilic attack towards methylglyoxal synthesized following the glycolytic switch in *Salmonella* undergoing oxidative stress. Despite the potential toxicity of the intermediates produced in this metabolic pathway, *Salmonella* engage the methylglyoxal pathway following peroxide stress. It has been argued that the methylglyoxal pathway is necessary in order to maintain increased glucose uptake during overflow metabolism. Our investigations, however, have shown that *ΔgloB Salmonella* support higher glucose uptake than wildtype cells. On the other hand, *ΔgloB Salmonella* exhibit lower aerobic respiration than wildtype cells. Thus, we conclude that the detoxification of the increased fluxes of methylglyoxal generated in response to the glycolytic switch enable optimal aerobic respiration. Importantly, the concomitant utilization of glycolysis and aerobic respiration supports resistance of *Salmonella* to oxidative killing by the phagocyte NADPH oxidase [22].

Our current and previous investigations have shown that *Salmonella* undergoing oxidative stress suffer a decrease in the concentration in reduced GSH. However, *Salmonella* undergoing peroxide stress do not experience concomitant increases in GSSG, as could be expected by the direct oxidation of GSH by $H_2O_2$. Moreover, a *ΔgshA Salmonella* strain deficient in the glutamylcysteine synthetase, the committed step in the biosynthesis of GSH, is hypersusceptible to $H_2O_2$ killing, whereas a *Δgor* strain deficient in the oxidoreductase that reduces GSSG to GSH is resistant to oxidative killing [14]. Together, this research suggests that the role glutathione plays in the antioxidant defenses of *Salmonella* is independent of canonical scavenging of $H_2O_2$. The hypersusceptibility of *ΔgloB* to peroxide stress offers an alternative mechanism for the key role GSH plays in the antioxidant defenses of *Salmonella*. We have uncovered that the glycolytic switch that follows the exposure of *Salmonella* to oxidative stress activates the methylglyoxal pathway. Our investigations suggest a novel role for glutathione in resistance of *Salmonella* to peroxide stress, whereby this low-molecular weight thiol quenches the highly toxic aldehyde intermediate synthesized during the conversion of dihydroxyacetone phosphate to pyruvate.

Our investigations have shown important roles of *gloB*-encoded glyoxalase II in the utilization of glucose by *Salmonella*. However, the *gloA*-encoded glyoxalase I appears to be dispensable. These findings may indicate that recycling of GSH from S-lactoylglutathione is more important than its formation as failing to do the former deprives the cell from the most abundant low-molecular-weight thiol. Other possibilities should also be considered. The apparent dispensability of *gloA* for the growth of *Salmonella* on glucose may reflect redundant mechanisms at this step of the methylglyoxal pathway. According to this idea, the *Salmonella* genome encodes two additional lactoylglutathione lyases [29]. The glyoxalase I encoded by *lgl* locus has been shown to help *Salmonella* detoxify methylglyoxal and grow in nutrient rich media [30]. Alternatively, glyoxalase II may have additional roles in resolving aldehyde-GSH adducts produced in metabolic pathways other than the methylglyoxal pathway. Regardless, *Salmonella* depends on functional *mgsA*, *gloA* and *gloB* genes for resistance to the oxidative killing emanating from the phagocyte NADPH oxidase.

In summary, our investigations have revealed critical roles for the methylglyoxal pathway for both the utilization of glucose and resistance of *Salmonella* to oxidative stress emanating from the phagocyte NADPH oxidase. The methylglyoxal pathway must be added to the pentose phosphate pathway, glycolysis and fermentation as key metabolic adaptations of *Salmonella* to oxidative stress [20,21,31–33].

## Materials and methods

### Ethics statement

This study was performed in accordance with the recommendations in the Guide for the Care and Use of Laboratory Animals of the National Institutes of Health. All animals were handled

in accordance with the Guide for the Care and Use of Laboratory Animals, following the approved Institutional Animal Care and Use Committee (IACUC) protocol 00059 of the University of Colorado School of Medicine (Assurance Number A3269-01), an AAALAC Accredited Institution.

## Bacterial strains, plasmids and growth conditions

The *Escherichia coli* (*E. coli*) strains DH5α and BL21(λDE3) pLysS were grown in Luria–Bertani (LB) broth at 37˚C in a shaking incubator or LB-agar at 37˚C. *Salmonella enterica* serovar *Typhimurium* strain 14028s (ATCC, Manassas, VA) and its mutant derivatives were grown in LB broth or MOPS minimal medium [40 mM MOPS buffer, 4 mM Tricine, 2 mM $K_2HPO_4$, 10 μM $FeSO_4 \cdot 7H_2O$, 9.5 mM $NH_4Cl$, 276 μM $K_2SO_4$, 500 nM $CaCl_2$, 50 mM NaCl, 525 μM $MgCl_2$, 2.9 nM $(NH_4)_6Mo7O_{24} \cdot 4H_2O$, 400 nM $H_3BO_3$, 30 nM $CoCl_2$, 9.6 nM $CuSO_4$, 80.8 nM $MnCl_2$, and 9.74 nM $ZnSO_4$; pH 7.2] supplemented with 0.4% D-glucose at 37˚C in a shaking incubator. Ampicillin (100 μg/ml), kanamycin (50 μg/ml), chloramphenicol (20 μg/mL) and tetracycline (20 μg/mL) were used where appropriate. Strains and plasmids used in the study are listed in **Tables A and B in S1 Text**.

## Construction of *Salmonella* mutants

λ-Red homologous recombination system was used to construct deletion mutants in *Salmonella* [34]. Specifically, the chloramphenicol cassette from the pKD13 plasmid was PCR amplified using primers with a 5' -end overhang homologous to the bases following the ATG start site and the bases preceding the stop codon of *mgsA*, *gloA* and *gloB* genes (**Table C in S1 Text**). The PCR products were gel purified, and electroporated into *Salmonella* expressing the λ Red recombinase from the plasmid pKD46 or pTP233. Transformants were selected on LB plates containing 50 μg/ml kanamycin. The mutants were confirmed by PCR and sequencing.

## Growth kinetics

*Salmonella* strains were grown overnight in LB broth at 37˚C in a shaker incubator. Stationary phase cultures were pelleted down and resuspended in MOPS-GLC minimal medium, pH 7.2, to an $OD_{600}$ of 1. The specimens were sub-cultured 1:100 into fresh MOPS-GLC minimal medium, pH 7.2, and grown at 37˚C with shaking at 37˚C. Where indicated, 0.4% D-glucose or 0.4% Casamino acids was added to MOPS minimal medium, pH 7.2. Bacterial growth was followed by recording $OD_{600}$ values every hour for 7–8 h at 37˚C in an aerobic shaking incubator.

## Intrabacterial redox potential

Intrabacterial redox potential was determined by fluorescent measurement for roGFP2 in *Salmonella* as described [35]. The plasmid pFPV25, that encodes and constitutively expresses roGFP2, was electroporated into wild-type *Salmonella* strain 14028s and Δ*greAB* mutant *Salmonella*. Overnight bacterial cultures were subcultured 1:100 in MOPS-GLC medium, pH 7.2 (without antibiotics) and grown at 37˚C with shaking to an $OD_{600}$ of 0.25. Culture aliquots (3 ml) were left untreated or treated with 400 μM $H_2O_2$ for 1 min at 37˚C before fluorescence measurement at excitation wavelengths of 405 and 480 nm (roGFP2$_{ox}$ and roGFP2$_{red}$, respectively). Emission was read at 510 nm in both instances. All values were normalized to the ratios obtained for fully oxidized and fully reduced cultures of the respective strains 1 min after treatment with 50 mM $H_2O_2$ or 10 mM DTT, respectively.

## Glutathione measurements

The concentrations of glutathione were estimated by the GSH recycling method described by Baker et al (1990) with modifications [20]. Briefly, *Salmonella* were grown in MOPS-GLC minimal medium, pH 7.2 at 37˚C to an $OD_{600}$ of 0.25. Where indicated, cells were treated with 400 μM $H_2O_2$ for 30 min. Cells were harvested by centrifugation at 10,000 rpm for 3 min. Bacterial pellets targeted for GSH and GSSG determinations were resuspended in 250 μl of 20 mM EDTA, pH 8.0, and in 20 mM EDTA, pH 8.0 containing 2 mM N-ethylmaleimide, respectively. The bacteria were lysed by sonication, and the cytoplasmic proteins in the specimens were precipitated upon the addition of 250 μl of 10% $HClO_4$. The samples were neutralized with 93.5 μl of 5 M KOH. The samples were freeze/thawed and cleared lysates were obtained by centrifugation at 12,000 rpm for 10 min. Reaction buffer was freshly prepared by mixing 2.8 ml of 1 mM 5,5′-dithiobis (2-nitro-benzoic acid), 3.75 ml of 1 mM NADPH, 5.85 ml of 100 mM NaH2PO4, 5 mM EDTA phosphate-EDTA buffer, and 20 U of glutathione reductase (Sigma-Aldrich). Hundred μl of reaction buffer was immediately added to 50 μl of sample in a 96-well microtiter plate. After a 5 min incubation, the reactions were read at $A_{412}$ in a spectrophotometer (Versamax Micrioplate Reader, Molecular Devices, USA). The concentration of GSH in the samples was estimated by regression analysis of known standards.

## Aldehyde and D-lactate measurement

Aldehyde and D-lactate production were measured by the Aldehyde Assay Kit and D-Lactate Assay Kit (Sigma-Aldrich), respectively, as per manufacturer's instructions. Briefly, *Salmonella* were grown to an $OD_{600}$ of 0.25 at 37˚C in a shaker incubator in MOPS-GLC medium, pH 7.2. Bacterial cells were sonicated in 200 μl of ice-cold lysis buffer (25mM Tris-HCl; pH 8.0 and 100 mM NaCl). Insoluble material was removed by centrifugation at 13,000 g for 10 min at 4˚C. Soluble cytoplasmic extracts were estimated in samples standardized to equal amount of total protein. For aldehyde measurement, 50 μl of the master reaction mix were added to 50 μl of samples. Aldehydes in a sample are reacted with 3-methyl-2-benzothiazolinone hydrazine (MBTH) at room temperature for 15–20 min. The blue color the MBTH-acetaldehyde complex formed was measured at 620 nm. The total concentration of aldehyde in a sample was determined from an aldehyde standard curve. For D-Lactate measurement, 80 μl of reaction mix were added to 20 μl containing lactate dehydrogenase and NAD+. NADH produced in the reaction reduces MTT to the formazan chromogen, which was measured at 565 nm. The total concentration of D-lactate in a sample were determined from the D-lactate standard curve.

## Glucose uptake

Glucose in the culture media was measured spectrophotometrically at 630 nm after derivatization with o-toluidine. Briefly, *Salmonella* strains grown in MOPS-GLC minimum medium, pH 7.2, at 37˚C to $OD_{600}$ of 0.25 were harvested. The cells were resuspended in fresh MOPS-GLC media, pH 7.2, and grown for 2 h at 37˚C. Some of the samples were treated with 400 μM $H_2O_2$. Culture supernatants were collected after 2 h and stored at -20˚C until further use. Five microliter of culture supernatants and known glucose standards mixed with 500 μl o-toluidine reagent were incubated at 100˚C for 8 min. Reaction mixtures were cooled down in ice for 4 min and the OD was recorded at 630 nm. Glucose concentration was calculated by regression analysis of known glucose standards.

## NADH and NAD⁺ measurements

Intracellular NADH/NAD$^+$ measurements were performed according as described [22]. Briefly, *Salmonella* were grown in MOPS-GLC minimum medium, pH 7.2, at 37˚C to an

OD$_{600}$ of 0.25. Where indicated, cells were treated for 30 min with 400 μM H$_2$O$_2$. NADH and NAD$^+$ were extracted from bacterial pellets with 120 μl of 0.2 M NaOH and 0.2 M HCl, respectively. The samples were neutralized by adding equal volumes of 0.2 M HCl and 0.2 M NaOH, respectively. 10 μl of the extracts were added to 90 μl of reaction buffer containing 200 mM bicine, pH 8.0, 8 mM EDTA, 3.2 mM phenazine methosulfate, 0.84 mM 3-(4,5-dimethylthiazol- 2-yl)−2,5- diphenyltetrazolium bromide, 20% ethanol and 0.4 μg alcohol dehydrogenase. Specimens were analyzed spectrophotometrically at 570 nm. NADH and NAD$^+$ were calculated by regression analysis of known standards. The results were normalized to OD$_{600}$.

## O$_2$ consumption

Consumption of O$_2$ was measured using an ISO-OXY-2 O$_2$ sensor attached to an APOLLO-4000 free radical analyzer (World Precision Instruments, Inc., Sarasota, FL) as described [22]. Briefly, *Salmonella* strains were grown aerobically to OD$_{600}$ of 0.25 in MOPS-GLC minimum medium, pH 7.2, at 37˚C. Three milliliter cultures were rapidly withdrawn and vortexed for one minute. O$_2$ concentration was recorded immediately for 150 seconds. A two-point calibration for 0 and 21% O$_2$ was measured as per manufacturer's instructions.

## Bone marrow-derived macrophages

Bone marrow cells were harvested in DMEM$^+$ medium [DMEM containing 10% fetal bovine serum (Thermo Fisher Scientific, Grand Island, NY), 2 mM glutamate, and 100 U/ml penicillin and 1 μg/mL streptomycin] from femurs of 8 week-old C57BL/6 or *cybb$^{-/-}$* mice. Cells were centrifuged at 500 g for 5 min, and the red blood cells were lysed in 1 ml of ACK lysis buffer for 1 min. The cells were mixed with 9 ml of DMEM$^+$ medium, and the specimens were centrifuged at 500 g for 5 min. The cells were resuspended in L cell-conditioned media (i.e., DMEM medium supplemented with 10% FBS, 5% horse serum, 1 mM sodium pyruvate and 20% L cell media prepared after culture of ATCC CCL1 929 L cells clone NCTC for 12 days at 37˚C in a 5% CO$_2$ incubator). Bone marrow cells were seeded at a concentration of 10$^7$ cells per 150-mm plate (Corning, St. Louis, MO) and grown for 4 days in a 5% CO$_2$ incubator at 37˚C. The cells were rinsed with prewarmed PBS, and the specimens were incubated with fresh L cell-conditioned media for 3 days. The resulting bone marrow cells were then adjusted to a final concentration of 3.5×10$^6$ cells/ml in DMEM$^+$ media, and 100 μl were added per well of a 96-well plate for the survival assays.

## Survival of *Salmonella* in bone marrow-derived macrophages

Bone marrow-derived macrophages that had been growing for 20 h in DMEM$^+$ medium were washed once with RPMI$^+$ medium (Sigma, St. Louis, MO) [RPMI medium supplemented with 10% heat-inactivated fetal bovine serum (Thermo Fisher Scientific, Grand Island, NY), 1 mM sodium pyruvate (Thermo Fisher Scientific, Grand Island, NY), 2 mM L-glutamine (Thermo Fisher Scientific, Grand Island, NY) and 20 mM HEPES (Thermo Fisher Scientific, Grand Island, NY). The cells were infected at an MOI of 2 with *Salmonella* grown overnight in LB broth. The specimens were centrifuged at 3000 g for 1 min, and the plates were incubated for 25 min at 37˚C in a 5% CO$_2$ incubator. *Salmonella*-infected bone marrow-derived macrophages were lysed with 0.1% Triton X-100 after 25 min of challenge to establish the levels of infection (i.e., t = 0). The medium was replaced with fresh RPMI$^+$ medium containing 10 μg/ml gentamicin, and the specimens were incubated for 2 or 18 h. At these time points, the specimens were treated with 0.1% Triton X-100, and the surviving bacteria were enumerated onto LB agar. The percent of *Salmonella* surviving in bone marrow-derived macrophages was calculated by comparing the CFU recovered 2 and 18 h after infection to the burden at time zero.

### Animal studies

Six to eight-week-old immunocompetent C57BL/6, and immunodeficient *cybb*$^{-/-}$ mice deficient in the gp91$^{phox}$ subunit of the phagocyte NADPH oxidase, respectively, were inoculated intraperitoneally with ~100 CFU of *Salmonella* grown overnight in LB broth at 37°C in a shaking incubator. Mouse survival was monitored over 14 days. The bacterial burden was quantified in livers and spleens 3 days post infection by plating onto LB agar containing the appropriate antibiotics. Competitive index was calculated as (strain 1/ strain 2)$_{output}$ / (strain 1/ strain 2)$_{input}$. The data are representative of two to three independent experiments. All mice experiments were conducted according to protocols approved by the Institutional Animal Care and Use Committee at the University of Colorado School of Medicine.

### Statistical analysis

Statistical analyses were performed using GraphPad Prism 5.0 software. One-way and two-way ANOVA, *t*-tests and logrank tests were used. Results were determined to be significant when $p < 0.05$.

## Supporting information

**S1 Text. Table A in S1 Text.** Bacteria used in this study. Table B in S1 Text. Plasmids used in this study. Table C in S1 Text. Oligonucleotides used in this study. Fig A in S1 Text. Schematic representation and structures of metabolites of methylglyoxal pathway in *Salmonella*. Fig B in S1 Text. Transcription of methylglyoxal pathway genes. Fig C in S1 Text. Growth of Δ*gloB Salmonella* in glucose medium after complementation. Fig D in S1 Text. Effect of peroxide on *Salmonella*. Fig E in S1 Text. Glutathione buffering capacity in *Salmonella* lacking glyoxalase I. Fig F in S1 Text. Nicotinamide adenine dinucleotide content in Δ*gloB Salmonella* undergoing oxidative stress.
(PDF)

## Acknowledgments

We thank Dr. Jessica Jones-Carson for kindly providing the mice.

## Author Contributions

**Conceptualization:** Sashi Kant, Andres Vazquez-Torres.

**Formal analysis:** Sashi Kant, Andres Vazquez-Torres.

**Funding acquisition:** Andres Vazquez-Torres.

**Investigation:** Sashi Kant, Lin Liu.

**Methodology:** Sashi Kant, Lin Liu.

**Supervision:** Andres Vazquez-Torres.

**Validation:** Lin Liu.

**Writing – original draft:** Andres Vazquez-Torres.

**Writing – review & editing:** Sashi Kant.

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
