## [Decision Letter · Decision Letter 0]

5 Apr 2023

Dear Dr. Vazquez-Torres,

Thank you very much for submitting your manuscript "The methylglyoxal pathway is a sink for glutathione in Salmonella experiencing oxidative stress" for consideration at PLOS Pathogens. As with all papers reviewed by the journal, your manuscript was reviewed by members of the editorial board and by several independent reviewers. The reviewers appreciated the attention to an important topic. Based on the reviews, we are likely to accept this manuscript for publication, providing that you modify the manuscript according to the review recommendations.

Sincerely,

Andreas J Baumler

Academic Editor

PLOS Pathogens

Nina Salama

Section Editor

PLOS Pathogens

Kasturi Haldar

Editor-in-Chief

PLOS Pathogens

orcid.org/0000-0001-5065-158X

Michael Malim

Editor-in-Chief

PLOS Pathogens

orcid.org/0000-0002-7699-2064

Reviewer Comments (if any, and for reference):

Reviewer's Responses to Questions

**Part I - Summary**

Reviewer #1: In the submitted manuscript entitled “The methylglyoxal pathway is a sink for glutathione in Salmonella experiencing oxidative stress”, the authors investigate and demonstrate the role of the methylglyoxal pathway and the importance of the glutathione recycling step in that pathway for the oxidative stress resistance. The authors show here that the addition of H202 alters the redox potential in Salmonella, but also activates the methylglyoxal pathway. The methylglyoxal pathway supports the growth of Salmonella on glucose, but also helps Salmonella to resist to H202. Indeed, the authors nicely demonstrate that the gloB mutant in Salmonella, which is unable to recycle GSH, is more sensitive to H202. By showing that the decrease of GSH concentration in presence of H202 is not accompanied by an increase of the oxidized form of GSH (GSSG) as it would be the case in the canonical scavenging of H2O2, the authors highlight here a new mode of action for the glutathione in the oxidative stress resistance in Salmonella.

The manuscript is well written, with an introduction addressing the major topics of the manuscript, with clear figures that are easy to understand, and with a discussion addressing the main results and the still-open questions. While the overall experimental strategy used is good, some additional experiments and/or explanations could be added to improve the manuscript in order to be considered for publication.

Reviewer #2: The manuscript entitled “The methylglyoxal pathway is a sink for glutathione in Salmonella experiencing oxidative stress” by Kant, S. et al. describes the effects of peroxide on various metabolic pathways in Salmonella, namely glycolysis, the methylglyoxal pathway and respiration. The authors show that upon application of peroxide stress, increases in aldehyde and D-lactate are consistent with rerouting of glycolytic flux through the methylglyoxal pathway. They also show that peroxide induces transcription of the gene encoding the first enzyme in the methylglyoxal pathway. The crux of the report is that peroxide lowers GSH/GSSG ratios by lowering GSH not having it directly oxidized to GSSG. They imply that the methylglyoxal pathway is a sink sequestering significant proportions of reduced GSH. They also show a role for this pathway in virulence in a Phox dependent fashion. The data are clearly laid out for the most part and the experiments are properly controlled. A few minor suggestions might improve the impact of the work overall:

Reviewer #3: In this manuscript, Kant, Liu, and Vazquez-Torres investigate the role of the methylglyoxal pathway of Salmonella as it undergoes conditions of oxidative stress, for example during infection. Peroxide treated cells show an increase in redox potential in their cytoplasm, however this is unlikely to be due oxidation of GSH to GSSG. Consistent with previous observations that glycolysis is critical for metabolism in the presence of ROS, the methyl glyoxal pathway appears to be highly active when cells are treated with peroxide (concentration of intermediates is increased). Genes whose products are involved in the methyl glyoxal pathway are upregulated in the presence of peroxide. A mutant lacking glyoxylase II (gloB) exhibited slower growth on glucose, likely because GSH is depleted from the cell through reaction with methylglyoxal. The gloB mutant is also less resistant to peroxide killing. The effect of peroxide exposure on GSH concentrations was negated in a gloB mutant, suggesting a link between the methylglyoxal pathway and peroxide stress in the presence of glucose. In bone marrow-derived macrophages and a mouse infection model, the gloB mutant is attenuated, and this attenuation rescued in Cybb deficiency.

A major conceptual advance of this work is clarifying the role of glutathione in mediating resistance of oxidative stress. Conventional wisdom holds that glutathione directly reacts with reactive oxygen species (peroxide), thus detoxifying these molecules. The data in this study suggest that glutathione is important during oxidative stress because it detoxifies aldehydes produced during overflow metabolism. This is a significant conceptual advance in our understanding of how pathogens deal with reactive oxygen stress.

The overall conclusions are justified by the data. The experimental data are convincing and well-controlled. There are some experiments that could enhance the study, such as a metabolic flux analysis of this pathway in vitro or measuring steady state concentrations of metabolites of interest (glyoxylate, S-lactoylglutathione). On the off-chance that the authors had already generated some data in this direction, it should be added to the current study. However, if the authors have no additional data generated at this point, I think this is fine (it would likely be confirmatory anyway).

**Part II – Major Issues: Key Experiments Required for Acceptance**

Reviewer #1: N/A

Reviewer #2: 1. The depletion of GSH in the gloB mutant likely results in oxidation of proteins and enzymes in the ETC. This would explain the reduced oxygen consumption of the gloB mutant as well as the enhanced glucose uptake. This could be verified by showing increased lactate/acetate production of the mutant, which was shown to have a growth defect in glucose containing media.

2. Figures 1, 4 and 5B: The scale for the GSSG levels should be changed so we can appreciate the levels in the different conditions.

3. Chemical structures in Figure 2 would be ideal.

4. Figure 3C should reflect the level of transcript not just fold induction. Do gloA and gloB show low induction because they are already abundantly expressed?

5. Lines 131, 151 and throughout. The use of “methylglyoxal mutants” is improper. Perhaps “mutants in genes encoding enzymes of the methylglyoxal pathway” or “methylglyoxal pathway mutants”.

6. Lines 77 and 236. Another word other than “cranked up” should be used for increased flux.

7. Line 238-39. Reference 28 is not appropriate to support this statement. There are several reviews on phosphosugar detoxification in bacteria.

Reviewer #3: (No Response)

**Part III – Minor Issues: Editorial and Data Presentation Modifications**

Reviewer #1: - Did the authors evaluate the concentration of GSH in a gloA mutant in the presence or in the absence of H202 stress ? Because a gloA mutant should be as defective as a gloB mutant for the GSH recycling as the GSH is linked in the hemithioacetal molecule after the spontaneous reaction between GSH and methylglyoxal, before the intervention of gloA and gloB enzymes. That would not change the conclusion about the fact that the gloA mutant is able to grow on glucose because of other possible redundant proteins, but that would confirm that gloA is as important as gloB in this new H202 resistance mechanism. Depending on the results, the sentence lines 137/138 could be nuanced.

This experiment could be interesting especially because there is a different phenotype in WT BMDMs (fig 7 A) between the gloA / mgs mutants and the gloB mutant, phenotype which is restored at the same level in Cybb-deficient BMDMs. This difference could be discussed as it is not so obvious why gloA and gloB have a different phenotype in WT BMDM.

- Did the authors try to measure the concentration of the methylglyoxal molecule itself in Salmonella in the presence or in the absence of H2O2 using a kit for example?

- Fig 7A: The method used to evaluate the bacterial burden in BMDMs after infection seems different between the time 0 and times 2 and 18 hours. Indeed, gentamycin treatment has been used only for T2 and T18, but not for T0. How to be sure that the bacterial burden at T0 correspond to the intracellular bacteria only? Did the authors washed several times the wells with PBS? Even if the results are expressed in percentage, using the same method for all timings will maybe avoid this high difference of bacterial burden between T0 and T2h. It could also be nice to show the results as “CFU/mL” with the absolute values for each timing (T0, T2 and T18), and not using a percentage.

- Fig 7B and C: Did the authors weight the organs before plating? That would allow to express the CFU per gram of organ, normalizing then all the data. If not, could the authors at least mention that they did (or not) observed any splenomegaly or hepatomegaly?

- The GSH and its canonical mode of action to scavenger H202 should be briefly introduced and described in the introduction as the experiments looked at the concentrations and ratio of GSH/GSSG in order to highlight another, new mode of action of GSH in the peroxide resistance.

- The protocol for the qRT-PCR done for the Fig 3C is missing in the “Materials and Methods” section. Please add the protocol in this section and explain the analysis method used.

-------------

- Please, go through the manuscript to check that all the abbreviations are defined the first time they are used. For examples: “MOPS” (line 95), “Reactive oxygen species (ROS)” (line 62), glutathione (GSH) (line 84), “MOPS minimum medium containing glucose as sole carbon source (MOPS-GLC) (line 96), “DTT” (line 96), “MTT” (line 355), …

- Instead of saying “loss of GSH” (line 112), the authors should mention the “decrease of GSH concentration” as GSH is still present in the condition where Salmonella is treated with H202 (Fig 1B).

- The authors should modify the following sentence “Reaction of methylglyoxal with GSH forms the S-lactoylglutathione conjugate” (line 116) as the first product formed after the reaction between methylglyoxal and GSH is the hemithioacetal, as showed in Fig 2.

- The “in PBS” (line 148) from “Salmonella grown to stationary phase in LB broth when exposed to H2O2 in PBS“ is a bit confusing. It should be either removed as it is never used somewhere else or clarified if the authors want to say that H2O2 is diluted in PBS. Also, if it is the case, it should be mentioned in the “Materials and Methods” section.

- Fig 1B, 5B and 5C: the scale of the Y axis should be modified in order to visualize that there is indeed no significant difference (Fig 1B and Fig 5C) or a significant difference (Fig 5B) in the GSSG concentration.

- Except if it is a double direction reaction, which does not appear on the Fig 2, the methylglyoxal cannot be detoxified by the methylglyoxal synthase (lines 240-241), as this enzyme is the one forming the methylglyoxal from DHAP. Please correct/precise.

- The use of the “greAB” mutant of Salmonella is described in the “Materials and Methods” section (line 318), but then never mentioned in the manuscript. Please, either modify the “Materials and Methods” or clarify the use of this mutant in the “Results” section.

- “Salmonella” should be in italic (line 598).

Reviewer #2: See above

Reviewer #3: Line 49: …“that is activated as overflow metabolism is favored during periods of oxidative stress” Run-on sentence? This is difficult to understand.

Reference 23: A more up-to-date review could be cited here, PMID: 34585982

Line 143: If I understand this correctly, the idea for explaining the gloB mutant phenotype is: Methylglyoxal, produced in large quantities from glucose degradation, reacts and thus depletes GSH. In the gloB mutant, S-lactoylglutathione accumulates, and S-lactoylglutathione acts as a GSH sink. It might be helpful for the reader to point that out as a main conclusion at the end of the paragraph.

PLOS authors have the option to publish the peer review history of their article (what does this mean?). If published, this will include your full peer review and any attached files.

Reviewer #1: No

Reviewer #2: No

Reviewer #3: No

Figure Files:

Data Requirements:

Reproducibility:

References:

---

## [Editor Report · Decision Letter 1]

23 May 2023

Dear Dr. Vazquez-Torres,

We are pleased to inform you that your manuscript 'The methylglyoxal pathway is a sink for glutathione in Salmonella experiencing oxidative stress' has been provisionally accepted for publication in PLOS Pathogens.

Best regards,

Andreas J Baumler

Academic Editor

PLOS Pathogens

Nina Salama

Section Editor

PLOS Pathogens

Kasturi Haldar

Editor-in-Chief

PLOS Pathogens

orcid.org/0000-0001-5065-158X

Michael Malim

Editor-in-Chief

PLOS Pathogens

orcid.org/0000-0002-7699-2064
---

## [Editor Report · Acceptance letter]

30 May 2023

Dear Dr. Vazquez-Torres,

We are delighted to inform you that your manuscript, "The methylglyoxal pathway is a sink for glutathione in Salmonella experiencing oxidative stress," has been formally accepted for publication in PLOS Pathogens.

Best regards,

Kasturi Haldar

Editor-in-Chief

PLOS Pathogens

orcid.org/0000-0001-5065-158X

Michael Malim

Editor-in-Chief

PLOS Pathogens

orcid.org/0000-0002-7699-2064